# The influence of preprocessing on text classification using a bag-of-words representation

**Yaakov HaCohen-Kerner**[ID]*, **Daniel Miller**[ID], **Yair Yigal**

Dept. of Computer Science, Jerusalem College of Technology - Lev Academic Center, Jerusalem, Israel

* kerner@jct.ac.il

**Data Availability Statement:** All R8 files are available from the http://ana.cachopo.org/datasets-for-single-label-text-categorization All WebKB files are available from the http://ana.cachopo.org/datasets-for-single-label-text-categorization are

## Abstract

Text classification (TC) is the task of automatically assigning documents to a fixed number of categories. TC is an important component in many text applications. Many of these applications perform preprocessing. There are different types of text preprocessing, e.g., conversion of uppercase letters into lowercase letters, HTML tag removal, stopword removal, punctuation mark removal, lemmatization, correction of common misspelled words, and reduction of replicated characters. We hypothesize that the application of different combinations of preprocessing methods can improve TC results. Therefore, we performed an extensive and systematic set of TC experiments (and this is our main research contribution) to explore the impact of all possible combinations of five/six basic preprocessing methods on four benchmark text corpora (and not samples of them) using three ML methods and training and test sets. The general conclusion (at least for the datasets verified) is that it is always advisable to perform an extensive and systematic variety of preprocessing methods combined with TC experiments because it contributes to improve TC accuracy. For all the tested datasets, there was always at least one combination of basic preprocessing methods that could be recommended to significantly improve the TC using a BOW representation. For three datasets, stopword removal was the only single preprocessing method that enabled a significant improvement compared to the baseline result using a bag of 1,000-word unigrams. For some of the datasets, there was minimal improvement when we removed HTML tags, performed spelling correction or removed punctuation marks, and reduced replicated characters. However, for the fourth dataset, the stopword removal was not beneficial. Instead, the conversion of uppercase letters into lowercase letters was the only single preprocessing method that demonstrated a significant improvement compared to the baseline result. The best result for this dataset was obtained when we performed spelling correction and conversion into lowercase letters. In general, for all the datasets processed, there was always at least one combination of basic preprocessing methods that could be recommended to improve the accuracy results when using a bag-of-words representation.

also avilable http://www.cs.cmu.edu/afs/cs/project/theo-20/www/data/ All WebKB files are also available from the http://ana.cachopo.org/datasets-for-single-label-text-categorization All SMS Spam Collection v.1 files are available from the http://www.dt.fee.unicamp.br/~tiago//smsspamcollection/ All Sentiment Labelled Sentences files are available from the https://archive.ics.uci.edu/ml/datasets/Sentiment+Labelled+Sentences# All Python 3 Spelling Corrector files are available from the https://github.com/phatpiglet/autocorrect/ All Cardoso-Cachopo(2007) files are available from the http://ana.cachopo.org/datasets-for-single-label-text-categorization All Keerthi et al. (2001) files are available from the http://doi.org/10.1162/089976601300014493.

**Funding:** The author(s) received no specific funding for this work.

**Competing interests:** The authors have declared that no competing interests exist.

# Introduction

Text classification (TC) is a supervised learning task that assigns natural language text documents to one (the typical case) or more predefined categories [1]. Classification algorithms typically use a supervised machine learning (ML) algorithm or a combination of several ML algorithms [2].

TC is an important component in many research domains including information extraction, information retrieval, text indexing, text mining, and word sense disambiguation [3]. There are two main types of TC: topic-based classification and stylistic classification. An example of a topic-based classification application is classifying news articles as Business-Finance, Lifestyle-Leisure, Science-Technology, and Sports [4]. An example of a stylistic classification application is classification based on different literary genres, e.g., action, comedy, crime, fantasy, historical, political, saga, and science fiction [5]. Whereas stylistic classification is typically performed using linguistic features such as quantitative features, orthographic features, part of speech (POS) tags, function words, and vocabulary richness features, topic-based classification is typically performed using unigrams and/or n-grams (for n > 2) [6–8].

The traditional model for topic-based TC is based on the bag-of-words (BOW) representation, which associates a text with a vector indicating the number of occurrences of each chosen word in the training corpus [2]. In a topic-based classification, ML methods such as Maximum Entropy (ME, Jaynes [9]), support vector machines (SVMs, Cortes and Vapnik [10]), Naive Bayes (NB, Heckerman [11]), and C4.5 decision tree induction [12,13] have been reported to use a BOW representation of hundreds or thousands of unigram words to achieve accuracies of 90% and greater for particular categories [1].

This study addresses topic-based classification based on the BOW representation. We hypothesize that the application of a selection of preprocessing methods can improve the accuracy results of different TC tasks. We investigate the impact of all possible combinations of five/six basic preprocessing methods on TC. We validate the proposed model on four benchmark corpora to determine what preprocessing combination(s) is (are) best suited to classification tasks.

The key contributions and novelties of this paper are (1) performing an extensive and systematic set of TC experiments using all possible combinations of five/six basic preprocessing methods on four benchmark text corpora (and not samples of them) using three ML methods and training and test sets; (2) presenting a general important conclusion about the importance of performing an extensive variety of preprocessing methods combined with TC experiments because it contributes to improve TC accuracy; and (3) presenting the specific findings that demonstrate what preprocessing combinations for what benchmark datasets significantly improve the classification results when using a BOW representation.

This paper is structured as follows. The **Material and Methods** section outlines the datasets, applied platform, applied supervised machine learning methods, preprocessing methods, and the chosen metric. In the **Preprocessing for Text Classification** section, we provide a detailed background for various common text preprocessing methods related to text classification. The **Examined Text Corpora** section introduces the four examined benchmark datasets. In the **Model** section, we introduce our classification Model. The **Experimental Results** section presents a wide variety of experimental results and their analysis. The **Summary, Conclusions, and Future Work** section summarizes the results and the main conclusion of this study and suggests a few future research proposals. Finally, the S1 Appendix presents examples of two full files for each of the four discussed datasets.

## Materials and methods

The four examined corpora in this study are benchmark datasets: WebKB, R8, SMS Spam Collection, and Sentiment Labelled Sentences. Details about them are found in the Examined Text

Corpora section. In this study, we applied three supervised ML methods: BN (Bayes Networks), SMO (a variant of SVM), and Random Forest (RF) using a bag of word unigrams. We used the WEKA platform with their default parameters. For each classification task, we used the experimenter mode in WEKA Version 3.9.1 with the following settings: Train (67%) / Test (33%) (data randomized) and the number of repetitions of each experiment was 10. We applied/implemented six basic preprocessing methods: C–spelling correction, L–converting uppercase letters into lowercase letters, H–HTML tag removal (relevant only for the WebKB dataset), P–punctuation mark removal, S–stopwords removal, and R–reduction of repeated characters. The chosen metric to evaluate the experimental results is the accuracy measure.

## Preprocessing for TC

Applying preprocessing on a certain dataset can improve its quality in general and for TC in particular. The preprocessing process can "clean" the dataset from "noise" (e.g., correction of spelling errors, reduction of replicated characters, and disambiguation of ambiguous acronyms). Furthermore, in some cases, application of preprocessing methods such as stopword removal, punctuation mark removal, word stemming, and word lemmatization can improve the dataset's quality for TC tasks.

There is a widespread variety of text preprocessing methods. Examples of basic types are conversion of uppercase letters into lowercase letters, correction of common misspelled words, HTML tag removal, punctuation mark removal, reduction of different sets of emoticon labels to a reduced set of wildcard characters, reduction of replicated characters, replacement of HTTP links to wildcard characters, stopword removal, and word stemming. Examples of more sophisticated preprocessing methods are word lemmatization, translation of common slang words into phrases that express the same ideas without using slang, and expansion of abbreviations.

A relatively small number of simple preprocessing methods (e.g., conversion of all the uppercase letters into lowercase letters and stopword removal) are performed by many TC systems. Nevertheless, most preprocessing methods are not performed by most TC systems. Furthermore, not all of the preprocessing methods are considered effective by all TC researchers. Some of them might even harm the classification results. For instance, Forman [14], in his study on feature selection metrics for TC, claimed that stopwords are ambiguous and occur sufficiently frequently as to not be discriminating for any particular class. However, HaCohen-Kerner et al. [15] demonstrated that the use of word unigrams including stopwords in the domain of Jewish law documents written in Hebrew–Aramaic lead to improved classification results compared to the results obtained using word unigrams excluding stopwords.

One of the explanations for the phenomenon that stopword removal harm the TC results is that the TF/TF-IDF values of the stopwords distribute differently in different categories improve style-based TC and in some cases even content-based TC. In addition, the contribution of use/non-use of stopwords depend on the corpus (even within the same language). For some corpora, various preprocessing methods might be irrelevant. For instance, HTML tag removal is not necessary on many SMS/Twitter corpora because of the relatively low prevalence of HTML tags.

Several studies analyzed the influence of various preprocessing methods on TC. A brief summary of many of these studies follows. Song et al. [16] examined 32 combinations of five preprocessing methods: stopword removal, word stemming, indexing with term frequency (TF), weighting with inverse document frequency (IDF), and normalization of each document feature vector to unit length. These combinations were applied to two benchmark datasets:

Reuters-21578 and 20 Newsgroups using a linear SVM and different lengths of a BOW representation. Their experimental results demonstrated that normalization to unit length can always significantly improve the effectiveness of text classifiers. Conversely, the impact of the other factors, e.g., stopword removal, word stemming, indexing, and weighting are rather minimal.

Lemmatization, stemming, and stopword removal were examined by [17] using only the multinomial NB classifier on two datasets: 8000 documents in English selected from Reuters Corpus Volume 1 dataset divided into six categories and 8000 Czech documents provided by Czech News Agency divided into five categories. They concluded that the best preprocessing approach for TC is to only apply stopword removal. Their experiments indicated that stopword removal improved the classification accuracy in most of the cases, although the results were not statistically significant. Further, lemmatization and stemming were more negative than positive for both languages.

The use of stopword removal, stemming, and different tokenization schemes on spam email filtering for two email corpora were analyzed in Méndez et al. [18]. They used three ML methods: NB, boosting trees, and SVM. Their main conclusion was that the performance of SVM is surprisingly effective when stemming and stopword removal are not used. One of the reasons is that some stopwords are rare in spam messages and should not be removed to improve the performance of spam filtering. Pomikálek and Rehurek [19] explored stopword removal, tokenization, and stemming by applying eight ML methods: SVM, NB, k-nearest neighbor (KNN, Fix and Hodges [20]), C4.5, neural networks, simple linear regression, voted perceptron, and RepTree [21] on limited versions of three datasets: Reuters 21578 (1800 documents), Newsgroups (6000 documents), and Springer (2700 documents). The authors concluded that stemming and stopword removal have minimal impact on the classification results.

Stopword removal, stemming, WordNet, and pruning were used for classification of MEDLINE documents in [22] using six ML methods: Sequential Minimal Optimization (SMO, Platt [23], Keerthi et al. [24]), Random Forest (RF, [25]), Bayes Networks (BayesNet, BN) [26], KNN, J48 (an improved variant of the C4.5 decision tree induction implemented in WEKA), and decision table [27]. Lowercase conversion, stopword removal, and tokenization were applied to all experiments without any comparison. Their experiments demonstrated that the application of pruning, stemming, and WordNet significantly reduces the number of attributes and improves the classification accuracy.

Srividhya and Anitha [28] evaluated four preprocessing methods: stopword removal, stemming, TF-IDF weighting [29], and document frequency on the Reuters 21578 dataset. Their main conclusions were as follows: (1) removal of stopwords can expand words and enhance the discrimination degree between documents and improve the classification performance, and (2) TF-IDF is required to create the index file from the resulting terms.

Clark and Araki [30] presented their system, called Casual English Conversion System (CECS). CECS deals with correction of errors and irregular words (which occur in social data) divided into eight categories as follows: (1) Abbreviation (shortform), (2) Abbreviation (acronym), (3) Typing error/ misspelling, (4) Punctuation omission/error, (5) Non-dictionary slang, (6) Wordplay, (7) Censor avoidance, and (8) Emoticons. CECS uses a manually compiled and verified database, which contained 1,043 entries. Each entry contain four columns: "error word" (the casual English item), "regular word" (the corresponding dictionary English item), "category" (the item's category), and "notes". The authors evaluated the performance of two open source English spell checkers on Twitter messages and measured the extent to which the accuracy of the spell checkers is improved by firstly preprocessing the texts with their system. The results showed that average errors per sentence decreased substantially, from roughly 15% to less than 5%.

Haddi et al. [31] investigated the role of text pre-processing in sentiment analysis of two online data sets of movie reviews (Dat-1400 and Dat-2000). They used a combination of different pre-processing methods (HTML tags removal, non-alphabetic signs removal, white space removal, abbreviation expansion, stemming, stop words removal, and negation handling) to reduce the noise in the text in addition to using chi-squared method to remove irrelevant features. They reported that sentiment analysis can be significantly improved using SVM, thousands of word unigrams, and appropriate pre-processing methods and feature selection. Their accuracy results are comparable to the accuracy results that can be achieved in topic classification, a much easier problem. On the Dat-2000 dataset they obtained an accuracy of 93.5% using 9,058 word unigrams.

Uysal and Gunal [32] studied the impact of preprocessing on TC using four preprocessing methods: tokenization, stopword removal, lowercase conversion, and stemming. Their examination was performed using all possible combinations of the preprocessing methods on four datasets: e-mails in Turkish, e-mails in English, news in Turkish, and news in English. They applied only the SVM ML method using feature sizes of 10-, 20-, 50-, 100-, 200-, 500-, 1,000-, and 2,000-word unigrams. Their main conclusion was that appropriate combinations of preprocessing tasks, depending on the domain and language, can provide a significant improvement in classification accuracy whereas inappropriate combinations can also degrade the accuracy. According to their experiments, lowercase conversion improves classification success in terms of accuracy and dimension reduction regardless of the domain and language. However, there is no unique combination of preprocessing tasks that provides successful classification results for every domain and language studied. Another finding is the importance of stopwords in contrast to many TC studies, which assume that stopwords are irrelevant.

Ayedh et al. [33] investigated the effect of three preprocessing methods (stopword removal, word stemming, and normalization of certain Arabic letters that have different forms in the same word to one form) on TC for an in-house corpus containing 32,620 news documents divided into ten categories downloaded from different Arabic news websites. In this study, three ML methods were applied: NB, kNN, and SVM. Experimental analysis revealed that preprocessing has a significant impact on the classification accuracy, especially with the complicated morphological structure of the Arabic language. Choosing appropriate combinations of preprocessing tasks provides significant improvement in the accuracy of TC depending on the feature size and the ML methods. The best result (a 96.74% micro-F1 value) was achieved by the SVM method using the combination of normalization and stemming.

Krouska et al. [34] performed various classification experiments using four ML methods (NB, SVM, KNN, and C4.5) with four preprocessing methods (TF-IDF weighting scheme, Stemming, Stop-words removal, and Tokenization). They did not applied all possible combinations of the preprocessing methods. The experiments were applied on three different datasets of tweets, one with no specific domain (the Stanford Twitter Sentiment Gold Standard (STS-Gold) dataset) and two datasets with specific topics (the Obama-McCain Debate (OMD) dataset and the Health Care Reform (HCR) dataset). The TF-IDF weighting scheme, stemming, and stop-words removal were applied as fixed options, while the experiments were with the tokenization and feature selection. The best accuracy results (92.67%, 92.59%, 91.94%) for the three datasets STS-Gold, OMD and HCR, respectively, have been obtained by NB using 720, 1074, and 1280 word 1-to-3-grams with InfoGain>0.

Jianqiang and Xiaolin [35] discussed the effects of text pre-processing methods on sentiment classification performance in two types of classification tasks, and summed up the classification performances of six pre-processing methods (replacing negative mentions (e.g., "won't" into "will not"), removing URL links, reverting words that contain repeated letters to their original English form, removing numbers, removing stop words, expanding acronyms to

their long forms) using two feature models (the word n-grams features model and the prior polarity score feature model) and four classifiers (SVM, NB, LR, RF) on five Twitter datasets (the Stanford Twitter Sentiment Test (STS-Test) dataset, SemEval2014-Task9 dataset, The Stanford Twitter Sentiment Gold (STS-Gold) dataset, The Sentiment Strength Twitter Dataset (SS-Twitter), The Sentiment Evaluation Dataset (SE-Twitter)). Their experimental results showed that the best preprocessing method was the replacement of negative mentions in the N-grams model. This method leads to a significant increase in the accuracy and F1-measure of almost all classifiers on all five datasets (the maximum improvement of accuracy is 8.23% using SVM and the maximum improvement of the F1-measure is 10.21% using RF on the SemEval2014 dataset). Expansion of acronyms showed significant improvement only for NB in the N-grams model (the accuracy and F1-measure obtained improvements of 6.85% and 6.08%, respectively). The other preprocessing methods (removal of URLs, the removal of stop words, reverting words, and the removal of numbers) minimally improve the classification accuracy. The NB and RF classifiers were found as more sensitive than LR and SVM classifiers when applying the pre-processing methods.

HaCohen-Kerner et al. [36] in their poster paper explored the impact of preprocessing methods on TC of three benchmark mental disorders datasets. They checked all possible combinations of the following six basic preprocessing methods (spelling correction, HTML tag removal, converting uppercase letters into lowercase letters, punctuation mark removal, reduction of repeated characters, and stopword removal) that were applied also in this research (Section 4). In the first dataset, the best result showed a significant improvement of about 28% over the baseline result using all the six preprocessing methods. In the two other datasets, several combinations of preprocessing methods showed only minimal improvement rates over the baseline results.To summarize the studies that have been presented above, both literally and as a summary in Table 1, there is no uniformity in the examined datasets, languages, preprocessing methods, ML methods, results, or conclusions. Most of the studies use a relatively small number of the following components: datasets, ML methods, preprocessing methods, and combinations of these. Moreover, portions of the conclusions of these studies seemingly contradict each other (e.g., stopword removal improves classification accuracy [15,17,21] or does not improve classification accuracy [16,31,37]).

Table 1 summarizes the attributes of twelve studies described in this section that addressed preprocessing for TC. The presented attributes are the dataset(s), ML methods, preprocessing methods, and best results and conclusions for each study.

The conclusions are not contradictory because they were derived from experiments on different datasets, possibly different languages, with different stopword lists, and different sizes of BOW representation.

Thus, we decided in this research to explore the influence of all possible combinations (in contrast to many of the previous studies) of five/six (more than the majority of the previous studies) basic preprocessing methods on TC for four benchmark corpora (and not samples of them) to determine what preprocessing combination(s) is (are) best suited to TC, if any. Our experiments are applied using training and test sets.

## Examined text corpora

The four examined corpora in this study are benchmark datasets: WebKB, R8, SMS Spam Collection, and Sentiment Labelled Sentences. Table 2 introduces general information about these four datasets. More details are provided after Table 2. Examples of two full files from each of these four datasets are given in the S1 Appendix.

Table 1. Attributes of studies addressing preprocessing for text classification.

| # | Study | # | Name | # | Name | # | Name | Best results and Conclusions |
|---|-------|---|------|---|------|---|------|------------------------------|
| | | **Dataset(s)** | | **ML methods** | | **Preprocessing methods** | | |
| 1 | Song et al. [16] | 2 | Reuters-21578 and 20 Newsgroups | 1 | Linear SVM | 5 | stopword removal, word stemming, indexing with TF, weighting with IDF, normalization of each feature vector to unit length | normalization to unit length always significantly improves |
| 2 | Méndez et al. [18] | 2 | email corpora | 3 | NB, Adaboost, SVM | 3 | stopword removal, stemming, and different tokenization schemes | SVM when stemming and stopword removal are not used achieves the best results |
| 3 | Toman et al. [17] | 2 | 8,000 English documents & 8,000 Czech documents | 1 | multinomial NB | 3 | stopword removal, different types of word normalization | stopword removal improves the accuracy, however, results are not statistically significant |
| 4 | Pomikálek and Rehurek [19] | 3 | limited versions of Reuters 21578 (1800 doc.), Newsgroups (6,000 doc.), Springer (2,700 doc.) | 8 | SVM. NB, kNN, C4.5, NN, Simple Linear Regression, Voted Perceptron, RepTree | 3 | stopword removal, tokenization, stemming | stemming and stopword removal have minimal impact |
| 5 | Gonçalves et al. [22] | 1 | a MEDLINE sample | 6 | SMO, RF, kNN, BN, J48, decision table | 4 | stopword removal, stemming, WordNet, pruning | lowercase conversion, stopword removal, and tokenization are not validated. pruning, stemming, and WordNet improve the accuracy |
| 6 | Srividhya and Anitha [28] | 1 | Reuters-21578 | N/A | N/A | 4 | stopword removal, stemming, TF-IDF, document frequency | removal of stopwords improves the system performance |
| 7 | Clark and Araki [30] | 1 | 100 sentences from Twitter messages | N/A | N/A | | Correction of errors and irregular words divided into 8 categories: (1) Abbreviation (shortform), (2) acronym, (3) Typing error/misspelling, (4) Punctuation omission/error, (5) Non-dictionary slang, (6) Wordplay, (7) Censor avoidance, and (8) Emoticons. | The performance of two open source English spell checkers on Twitter messages was improved by firstly preprocessing the texts with their system. The results showed that average errors per sentence decreased substantially, from roughly 15% to less than 5%. |
| 8 | Uysal and Gunal [32] | 4 | e-mails in Turkish/English, news in Turkish/English | 1 | SVM | 4 | tokenization, stopword removal, lowercase conversion, stemming | lowercase conversion improves classification whereas stop-words should not be removed |
| 9 | Ayedh et al. [33] | 1 | 32,620 Arabic news | 3 | NB, KNN, SVM | 3 | stopword removal, word stemming, normalization of certain Arabic letters to one form | SVM using normalization and stemming significantly improves the accuracy |
| 10 | Krouska et al. [34] | 3 | the Stanford Twitter Sentiment Gold Standard (STS-Gold) dataset, the Obama-McCain Debate (OMD) dataset, the Health Care Reform (HCR) dataset | 4 | NB, SVM, KNN, and C4.5 | 4 | TF-IDF weighting scheme, Stemming, Stop-words removal, and Tokenization | The TF-IDF weighting scheme, stemming, and stop-words removal were applied as fixed options, while the experiments were with the tokenization and feature selection. The best accuracy results (92.67%, 92.59%, 91.94%) for the three datasets STS-Gold, OMD and HCR, respectively, have been obtained by NB using 720, 1074, and 1280 word 1-to-3-grams with InfoGain>0. |

(*Continued*)

**Table 1.** (Continued)

| # | Study | # | Name | # | Name | # | Name | Best results and Conclusions |
|---|-------|---|------|---|------|---|------|------------------------------|
| | | | Dataset(s) | | ML methods | | Preprocessing methods | |
| 11 | Jianqiang and Xiaolin [35] | 5 | Twitter datasets: (the Stanford Twitter Sentiment Test (STS-Test), SemEval2014-Task9, The Stanford Twitter Sentiment Gold (STS-Gold), The Sentiment Strength Twitter (SS-Twitter), The Sentiment Evaluation (SE-Twitter) | 4 | SVM, NB, LR, RF | 6 | replacing negative mentions, removing URL links, reverting words that contain repeated letters to their original English form, removing numbers, removing stop words, expanding acronyms to their long forms | Replacement of negative mentions in the N-grams model leads to a significant increase (maximum accuracy improvement of 8.23% using SVM and maximum F1-measure improvement of 10.21% using RF on the SemEval2014 dataset). Expansion of acronyms showed significant improvement only for NB in the N-grams model (the accuracy and F1-measure obtained improvements of 6.85% and 6.08%, respectively). The other preprocessing methods minimally improve the classification accuracy. The NB and RF classifiers were found as more sensitive than LR and SVM classifiers when applying the pre-processing methods. |
| 12 | HaCohen-Kerner et al. [36] | 3 | Three mental disorder datasets: CLPsych 2015 CLPsych 2016 CLPsych 2017 | 3 | BN, SMO, RF | 6 | spelling correction (C), HTML tag removal (H), converting uppercase letters into lowercase letters (L), punctuation mark removal (P), reduction of repeated characters (R), and stopword removal (S) | In CLPsych-15, the best accuracy result (90.32) achieved by RF showed a significant improvement over the baseline using all the six preprocessing methods. In the other datasets, several combinations of preprocessing methods showed minimal improvement rates over the baseline results achieved by RF in CLPsych-16 (67.95 using CLS) and SMO in CLPsych-17 (67.63 using CL) |

The WebKB dataset (The 4 Universities Data Set) [32,38,39]) contains 4,199 documents. Each document contains around 10,358 characters that are divided into around 2,918 words. The documents are webpages collected by the World-Wide Knowledge Base project of the CMU text-learning group. These pages were manually classified into seven different classes: student, faculty, staff, department, course, project, and other. We worked with the original WebKB dataset and not with its preprocessed version where all the terms were converted to stems.

The R8 dataset is a single-labeled dataset derived from the multi-labeled Reuters-21578 dataset; its documents were selected from the Reuters newswire in 1987 and manually

**Table 2. General information about the examined datasets.**

| # | Dataset | # of documents | # of categories | # of words | # of characters | Avg. # of words per doc. | Avg. # of characters per doc. |
|---|---------|----------------|-----------------|------------|-----------------|--------------------------|-------------------------------|
| 1 | WebKB | 4,199 | 7 | 12,253,042 | 43,492,981 | 2,918.09 | 10,357.94 |
| 2 | R8 | 7,674 | 8 | 785,610 | 4,502,926 | 102.37 | 586.78 |
| 3 | SMS Spam Collection | 5,574 | 2 | 104,026 | 454,864 | 18.66 | 81.60 |
| 4 | Sentiment Labelled Sentences | 3,000 | 2 | 41,278 | 198,831 | 13.76 | 66.28 |

classified by personnel from Reuters Ltd. The documents contained in R8 are selected from 8 of the 10 most frequent classes of Reuters-21578. These R8 classes contain 7,674 documents with a single topic and all classes have at least one training and one test document. Each document contains around 587 characters that are divided into around 102 words.

The SMS Spam Collection v.1 [40] dataset (http://www.dt.fee.unicamp.br/~tiago//smsspamcollection/) is a public set of SMS labeled messages that have been collected for mobile phone spam research. This dataset contains 5,574 English, real, and non-encoded messages, tagged as being legitimate (ham) or spam. Each SMS message contains around 82 characters that are divided into around 19 words. The dataset is composed of four subsets: (1) 425 SMS spam messages manually extracted from the Grumbletext website; (2) 3,375 SMS randomly chosen ham messages of the NUS SMS Corpus (NSC), which is a dataset of approximately 10,000 legitimate messages collected for research at the Department of Computer Science at the National University of Singapore; (3) 450 SMS ham messages collected from Caroline Tagg's PhD Thesis [41]; and (4) the SMS Spam Corpus v.0.1 Big, which contains 1,002 SMS ham messages and 322 spam messages.

The Sentiment Labelled Sentences dataset (https://archive.ics.uci.edu/ml/datasets/Sentiment+Labelled+Sentences#) is provided by the University of California, Irvine. This dataset contains 3000 reviews/sentences composed of an equal number of reviews (1,000) from three websites (imdb.com, amazon.com, and yelp.com). Each document contains around 66 characters that are divided into around 14 words. These are reviews of products, movies, and restaurants. For each website, there exist 500 positive and 500 negative sentences. Each review in the dataset is tagged as positive or negative. This dataset was created for the study presented in [42].

## Model

Our approach is to compare the accuracy results obtained using the original files without any preprocessing to the results achieved using combinations of five/six preprocessing methods: **C**–spelling correction, **H**–HTML tag removal (for relevant datasets), **L**–converting uppercase letters into lowercase letters, **P**–punctuation mark removal, **R**–reduction of repeated characters, and **S**–stopword removal. These six preprocessing methods are relatively basic and/or common. Due to this, many TC systems applied at least part of these methods. Therefore, we decided to apply them. Furthermore, we consider all possible combinations of these five/six basic preprocessing methods, i.e., 31 ($2^5$–1) / 63 ($2^6$–1) nonempty combinations. The spelling correction is achieved using the Python 3 Spelling Corrector (https://github.com/phatpiglet/autocorrect/). The application of the S preprocessing method deletes all instances of 423 stopwords for English text (421 stopwords from Fox [43] plus the letters "x" and "z" that are not found in Fox [41], yet are included in many other stopword lists).

Various studies explored the selection of appropriate ML methods. Kotsiantis et al. [44] presented a review of classification techniques. Their main conclusion was that the key question when dealing with supervised ML classification is not whether a certain ML method is superior to others, but under which conditions a certain ML method can significantly outperform others for a given task.

Fernández-Delgado et al. [45] evaluated 179 ML methods arising from 17 ML method families on 121 data sets. The best results were achieved by (1) Parallel RF (implemented in R with caret), (2) RF in R tuned with caret, and (3) The LibSVM implementation of SVM in C with Gaussian kernel. Six RFs and five SVMs are included among the 20 best classifiers.

Krouska et al. [46] introduced a comparative analysis of five well-known ML methods: NB, SVM, KNN, Logistic Regression (LR, [47]), and C4.5 and a lexicon-based approach called SentiStrength (Available: http://sentistrength.wlv.ac.uk, last accessed on 2019 December 11.).

These five ML methods were chosen as the most representative of ML and lexicon-based methods and were tested using three datasets and two test models (percentage split and cross validation). The best results were obtained by NB and SVM regardless of datasets and test methods.

In this research, we applied three supervised ML methods: BN, SMO, and RF. SMO (a variant of SVM) and RF were among the best ML methods found by the studies mentioned above. BN was found by us as the best ML method out of four ML methods including RF for a topic-based classification task [14]. We used the WEKA platform with their default parameters [20,48]. For each TC task, we used the experimenter mode in WEKA Version 3.9.1 with the following settings: Train (67%) / Test (33%) (data randomized) and the number of repetitions of the experiment set to 10.

A brief description of these three selected ML methods follows: BN is a variant of a probabilistic statistical classification model that represents a set of random variables and their conditional dependencies via a directed acyclic graph [25]. SMO [22,23] is a variant of the SVM ML method [9]. The SMO method is an iterative method created to solve the optimization problem frequently found in SVM methods. SMO divides this problem into a series of the smallest possible sub-problems, which are then resolved analytically. RF is an ensemble learning method for classification and regression [24]. Ensemble methods use multiple learning algorithms to obtain improved predictive performance compared to what can be obtained from any of the constituent learning algorithms. RF operates by constructing a multitude of decision trees at training time and outputting classification for the case at hand. RF combines Breiman's "bagging" (Bootstrap aggregating) idea in [49] and random selection of features introduced by Ho [50] to construct a forest of decision trees.

As mentioned in the Introduction section, the traditional model for topic-based TC is based on the bag-of-words (BOW) representation and many ML methods have been reported to use a BOW representation of hundreds or thousands of unigram words to achieve accuracies of 90% and greater for various TC tasks. Therefore, we decided to apply the traditional and successful BOW model of 1,000 to 5,000 (in steps of 1,000) unigram words in order to check our hypothesis that the application of different combinations of preprocessing methods can improve TC results. In Fig 1, we present a flowchart of our classification model.

## Experimental results

To determine a reasonable number of word unigrams for the BOW presentation that we intend to apply, we performed TC experiments for each pair of dataset and ML method, using five different sets, containing the TF values of 1,000-, 2,000-, 3,000-, 4,000-, and 5,000-word unigrams. Tables 3–6 present the TC accuracy results using the TF values of 1,000-, 2,000-, 3,000-, 4,000-, and 5,000-word unigrams for the four examined benchmark datasets: WebKB, R8, SMS Spam Collection, and Sentiment Labelled Sentences. These tables refer to the experimental results that were performed without any normalization/preprocessing methods. These tables contain different annotations and emphases. The annotation v or * indicates that a specific result in a certain column is statistically better (v) or worse (*) than the baseline result (the result using 1,000-word unigrams). To compare the different results, we performed statistical tests using a corrected paired two-sided t-test with a confidence level of 95%. A number in italics represents the best accuracy result for each ML method (one per column) and a number in bold represents the best accuracy result in the table (i.e., the best accuracy result for the discussed dataset).

The main findings presented in Tables 3–6 are as follows: (1) the best accuracy results were achieved by the SMO method (1st place in three datasets and 2nd place in one dataset), and the second best accuracy results were obtained by the RF method (1st place in one dataset and

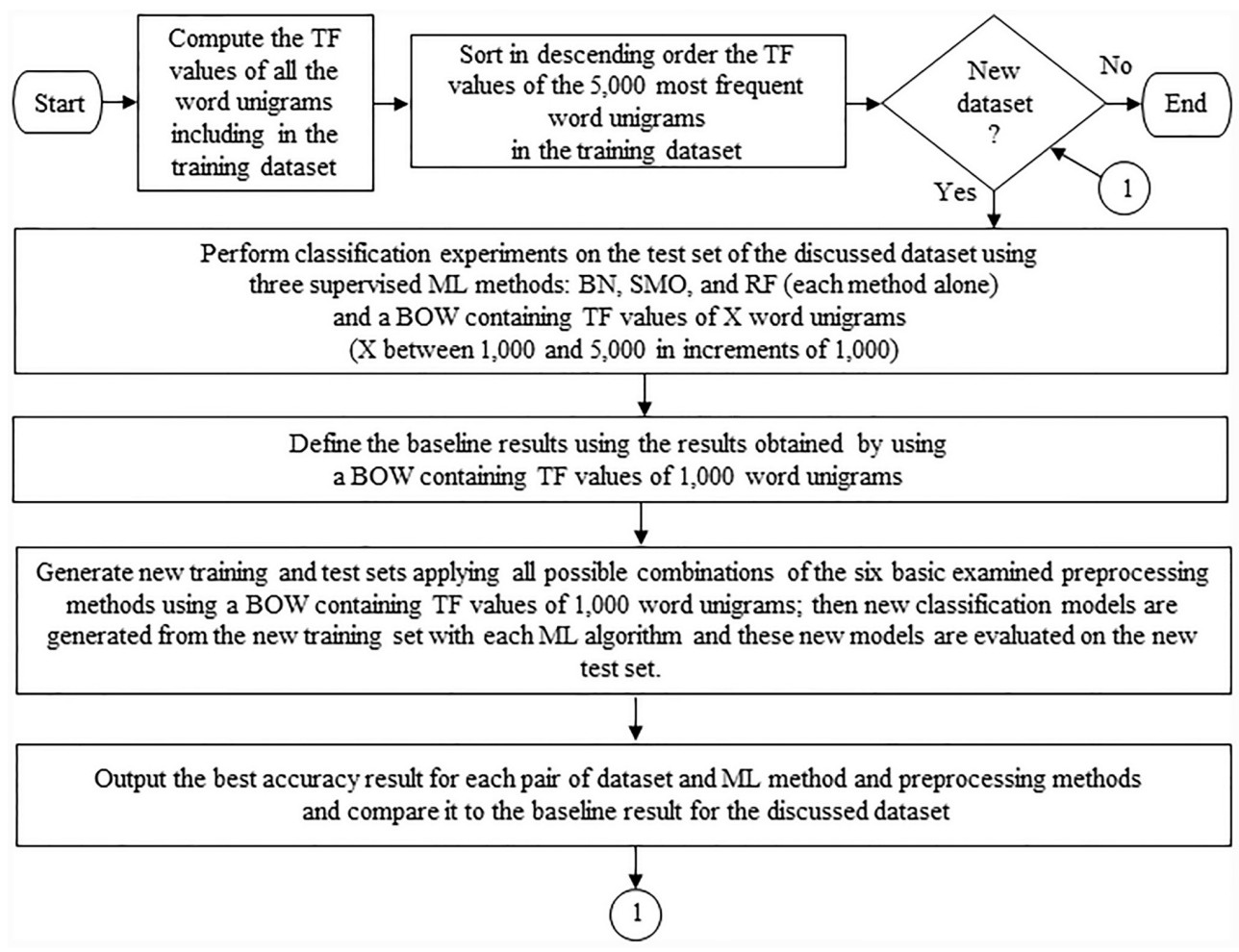

**Fig 1. Flowchart of the classification model.**

2nd place in three datasets); (2) one best accuracy result was obtained using 1,000-word unigrams for the WebKB dataset; two best accuracy results were obtained using 2,000-word unigrams for the R8 and SMS Spam Collection datasets; and one best accuracy result was obtained using 3,000-word unigrams for the Sentiment Labelled Sentences dataset (higher in 0.01 than the result for 2,000-word unigrams); and (3) for all datasets, neither the 4,000- nor 5,000-word unigrams achieved the best accuracy result.

In two datasets (R8 and SMS Spam Collection), the best accuracy result was achieved using 2,000-word unigrams; it was only marginally and not significantly better than the compatible

**Table 3. TC results for different numbers of word unigrams on the WebKB dataset.**

| # of Word Unigrams | BN | SMO | RF |
|---|---|---|---|
| 1000 | 86.01 | **94.10** | *93.99* |
| 2000 | 86.80v | 93.63 | 92.70* |
| 3000 | 87.29v | 93.50* | 91.55* |
| 4000 | 87.85v | 93.46* | 90.58* |
| 5000 | *88.28v* | 93.29* | 90.04* |

Table 4. TC results for different numbers of word unigrams on the R8 dataset.

| # of Word Unigrams | BN | SMO | RF |
|---|---|---|---|
| 1000 | 81.99 | 94.98 | *91.73* |
| 2000 | 82.51v | **95.10** | 90.31* |
| 3000 | 82.59v | 95.05 | 89.78* |
| 4000 | 82.60v | 94.89 | 89.23* |
| 5000 | *82.63v* | 94.70 | 88.97* |

accuracy results using 1,000-word unigrams. The use of 1,000-word unigrams instead of 2,000 saved a significant amount of time in our extensive set of experiments (64 preprocessing combinations for each pair of dataset and ML method). Thus, we decided to perform the TC experiments using the top 1,000 frequent word unigrams for each dataset.

Tables 7–10 present the TC accuracy results for the four examined datasets: WebKB, R8, SMS Spam Collection, and Sentiment Labelled Sentences using the top 1,000 frequent word unigrams for each dataset separately. As mentioned previously, we compared the results obtained using the original files without any preprocessing to the results achieved using all possible combinations of the following five/six basic preprocessing methods: C–spelling correction, L–lowercase, H–html tags (relevant only for the WebKB dataset), P–punctuation, S–stopwords, and R–repeated characters. We considered all 31 ($2^5-1$) / 63 ($2^6-1$) nonempty combinations of the five/six preprocessing methods in addition to the empty combination without the use of any preprocessing method.

Also in these tables, there are different annotations and emphases. The annotation v or * indicates that a specific result in a certain column is statistically better (v) or worse (*) than the baseline result (the result using 1,000-word unigrams) at the significance level of 0.05. A number in italics represents the best accuracy result for each ML method (one per column) and a number in bold represents the best accuracy result in the table (i.e., the best accuracy result for the discussed dataset).

Table 7 (TC for the WebKB dataset) indicates that for each one of the ML methods, there were several significant improvements. Regarding the impact of each single preprocessing method on the WebKB dataset, the S preprocessing (i.e., Stopword removal) was the only single preprocessing method that enabled significant improvements compared to the baseline result according to the accuracy measure. This was done using two ML methods: SMO (95.61% vs. 94.1%) and RF (95.04% vs. 93.99%) while the BN ML method showed an insignificant improvement. The C and H (spelling Correction, Html object removal) preprocessing methods enabled insignificant improvement for all three ML methods. The **R** (Reduction of repeated characters) preprocessing method enabled an insignificant improvement for two ML methods (BN and RF) and an insignificant decline for the SMO ML method. The **L** (i.e., converting uppercase letters into Lowercase letters) preprocessing methods showed insignificant

Table 5. TC results for different numbers of word unigrams on the SMS spam collection dataset.

| # of Word Unigrams | BN | SMO | RF |
|---|---|---|---|
| 1000 | 97.40 | 97.44 | *97.62* |
| 2000 | *97.54v* | **97.84** | 97.48 |
| 3000 | *97.54v* | 97.68 | 97.32 |
| 4000 | *97.54v* | 97.73 | 97.18 |
| 5000 | *97.54v* | 97.68 | 97.00* |

**Table 6. TC results for different numbers of word unigrams on the sentiment labelled sentences dataset.**

| # of Word Unigrams | BN | SMO | RF |
|---|---|---|---|
| 1000 | *65.76* | 73.74 | 75.99 |
| 2000 | 65.76 | 74.43v | 76.26v |
| 3000 | 65.76 | *74.88v* | **76.27v** |
| 4000 | 65.76 | 74.48v | 73.62v |
| 5000 | 65.76 | 74.78v | 73.99 |

declines for all three ML methods while the **P** (i.e., Punctuation mark removal) preprocessing method showed significant declines for all three ML methods.

Better results that are significant improvements compared to the baseline result have been obtained by various combinations. The best result 95.74% was obtained by the SMO ML method using the CHS combination (spelling correction, html object removal, and stopword removal), a combination of the only three preprocessing methods that showed improvements for all three ML methods. The S preprocessing method was a part of the most combinations that achieved significant improvements. This finding is not new and was also observed by [15,17,21]. It also fits the claim of Forman [13] that stopwords are ambiguous and occur sufficiently frequently as to not be discriminating for any particular class for TC. Secondary findings are the significant contributions of the H and C preprocessing methods to the S preprocessing method that enabled achieving the best results using SMO and RF.

Table 8 (TC for the R8 dataset) demonstrates similar findings to those of Table 7. Regarding the impact of each single preprocessing method on the R8 dataset, again, for each one of the ML methods, there are several significant improvements and again the S preprocessing was the only single preprocessing method that enabled a significant improvement compared to the baseline result; this time, for all three ML methods: SMO (95.75% vs. 94.98%), RF (94.46% vs. 91.73%), and BN (90.40% vs. 81.99%). The L and P preprocessing methods showed no change for all three ML methods while the C and R preprocessing methods showed insignificant declines for two ML methods and only one insignificant improvement for one ML method.

There were marginal significant improvements such as 95.75% using many combinations such as S (mentioned in the previous paragraph), LP, and PS. The S preprocessing method was again a part of the most combinations achieving significant improvements.

Table 9 (TC for the SMS Spam Collection dataset) demonstrates similar findings to those of Tables 7 and 8. Regarding the impact of each single preprocessing method on the SMS Spam Collection dataset, once more, the S preprocessing was the only single preprocessing method that enabled a significant improvement compared to the baseline result; this time, only for the RF method: 98.34% (the best result in Table 9) vs. 97.62% (the best baseline result). The R and L preprocessing methods showed two insignificant improvements and one insignificant decline. The C preprocessing method showed one insignificant improvement, one insignificant decline, and one no change. The P preprocessing method showed significant declines for all three ML methods (similar to the results in Table 7).

For this dataset (the SMS Spam Collection dataset), there were marginal improvements for each ML method; however, only two were significant: both were 98.34% for the RF method using the S and LP combinations.

Table 10 (TC for the Sentiment Labelled Sentences dataset) demonstrates totally different findings compared to those of the other datasets (Tables 7–9). Indeed, each one of the ML methods presents various significant improvements. Regarding the impact of each single preprocessing method on the Sentiment Labelled Sentences dataset, the only single preprocessing method that enabled a significant improvement compared to the baseline result was L

**Table 7. TC results for all possible combinations on the WebKB dataset.**

| # | Preprocessing(s) | BN | SMO | RF |
|---|---|---|---|---|
| 1 | None | 86.01 | 94.10 | 93.99 |
| 2 | C | 86.67 | 94.21 | 94.20 |
| 3 | H | 86.28 | 94.28 | 94.24 |
| 4 | L | 85.87 | 94.07 | 93.86 |
| 5 | P | 81.91* | 87.05* | 91.39* |
| 6 | R | 86.08 | 94.08 | 94.10 |
| 7 | S | 86.31 | 95.61v | 95.04v |
| 8 | CH | 87.53v | 94.30 | 94.58 |
| 9 | CL | 84.23* | 94.27 | 93.32 |
| 10 | CP | 84.89* | 92.78* | 92.64* |
| 11 | CR | 86.04 | 94.28 | 94.18 |
| 12 | CS | 84.13* | 94.34 | 94.04 |
| 13 | HR | 86.48 | 94.18 | 94.09 |
| 14 | HS | 86.37 | 95.63v | *95.19v* |
| 15 | LH | 85.98 | 93.98 | 94.07 |
| 16 | LP | 81.49* | 87.17* | 91.44* |
| 17 | LR | 86.14 | 94.01 | 93.98 |
| 18 | LS | 85.92 | 95.51v | 94.67 |
| 19 | PH | 81.18* | 86.24* | 91.21* |
| 20 | PR | 82.18* | 86.92* | 91.48* |
| 21 | PS | 82.87* | 88.96* | 92.17* |
| 22 | SR | 86.52 | 95.55v | 94.78v |
| 23 | CHL | 86.54 | 94.04 | 93.70 |
| 24 | CHP | 82.13* | 85.86* | 90.64* |
| 25 | CHR | 87.42v | 94.21 | 94.05 |
| 26 | CHS | *88.90v* | **95.74v** | 94.86v |
| 27 | CLP | 80.47* | 85.93* | 89.48* |
| 28 | CLR | 84.21* | 94.24 | 93.11* |
| 29 | CLS | 85.07 | 95.60v | 93.72 |
| 30 | CPR | 84.62* | 92.67* | 92.70* |
| 31 | CPS | 86.50 | 93.76 | 93.46 |
| 32 | CRS | 87.51v | 95.58v | 94.76 |
| 33 | HSR | 86.74 | 95.54v | 95.04v |
| 34 | LHR | 86.35 | 93.98 | 93.79 |
| 35 | LHS | 85.89 | 95.39v | 94.85 |
| 36 | LPH | 80.81* | 86.31* | 90.98* |
| 37 | LPR | 82.09* | 87.14* | 91.52* |
| 38 | LPS | 83.06* | 89.00* | 92.27* |
| 39 | LSR | 86.26 | 95.44v | 94.53 |
| 40 | PHR | 81.69* | 86.31* | 91.37* |
| 41 | PHS | 83.25* | 88.57* | 92.35* |
| 42 | PSR | 83.46* | 88.95* | 92.25* |
| 43 | CHLP | 82.09* | 85.83* | 90.14* |
| 44 | CHLR | 86.63 | 93.99 | 93.39 |
| 45 | CHLS | 87.48v | 95.53v | 94.06 |
| 46 | CHPR | 82.18* | 85.90* | 90.80* |
| 47 | CHPS | 85.03 | 87.63* | 91.39* |

(*Continued*)

**Table 7.** (Continued)

| # | Preprocessing(s) | BN | SMO | RF |
|---|---|---|---|---|
| 48 | **CHRS** | 89.05v | 95.66v | 94.85v |
| 49 | **CLPR** | 81.55* | 86.72* | 90.56* |
| 50 | **CLPS** | 82.49* | 87.85* | 90.86* |
| 51 | **CLRS** | 85.29 | 95.52v | 93.38 |
| 52 | **CPRS** | 86.65 | 93.72 | 93.20 |
| 53 | **LHRS** | 86.20 | 95.29v | 94.76v |
| 54 | **LPHR** | 81.34* | 86.29* | 91.08* |
| 55 | **LPHS** | 83.10* | 88.34* | 92.14* |
| 56 | **LPRS** | 83.81* | 88.87* | 92.26* |
| 57 | **PHSR** | 83.88* | 88.24* | 92.47* |
| 58 | **CHLPR** | 82.20* | 85.93* | 90.23* |
| 59 | **CHLPS** | 84.91 | 87.94* | 91.00* |
| 60 | **CHLRS** | 87.78v | 95.44v | 94.06 |
| 61 | **CHPRS** | 85.36 | 87.67* | 91.39* |
| 62 | **CLPRS** | 82.89* | 87.90* | 90.92* |
| 63 | **LHPRS** | 83.83* | 88.30* | 92.26* |
| 64 | **CHLPRS** | 85.31 | 87.82* | 90.73* |

(converting uppercase letters into lowercase letters). The L type presents significant improvements by RF (78.22% vs. 75.99%) and BN (69.26% vs. 65.76%) and an insignificant improvement by SMO. The C preprocessing method enabled insignificant improvement for all three ML methods. The R preprocessing method showed two insignificant improvements and one insignificant decline. The P preprocessing method showed insignificant declines for all three ML methods. Quite inversely to the three previous datasets, the S preprocessing method was the worst preprocessing method. The S preprocessing method presented significant declines for two ML methods (i.e., for BN 54.37% vs. a baseline of 65.76% and RF 73.19% vs. a baseline of 75.99%) and one insignificant decline for SMO.

The best result in Table 10, 78.78%v (a significant improvement of 2.79% from the baseline) was obtained by the RF method using CL. The majority of the improvement is because of the L preprocessing, which presents a significant improvement of 2.23% from the baseline; SMO, the second best ML method for the Sentiment Labelled Sentences dataset, demonstrates similar results to those obtained by the RF method. The best result, 75.81% was obtained using CL. The S preprocessing method did not appear in any of the combinations of all three ML methods that achieved significant improvements compared to the baseline results.

Possible explanations for the relatively low TC accuracy results and the surprising preprocessing results for the Sentiment Labelled Sentences dataset could be the fact that many reviews are implicit or indirect or complicated by positive and negative parts intertwined. Another possible explanation is the one presented in Toman et al. [16] that some stopwords are rare in these sentiment sentences (as in spam messages because they are both relatively short) and should not be removed to improve the classification accuracy.

Table 11 presents a summary of the three best accuracy results in descending order for each dataset.

## Analysis of the results for all datasets

In all four datasets, we have obtained significant improvements compared to the baseline results. For each dataset, the best combination(s) of preprocessing methods was(were)

**Table 8. TC results for all possible combinations on the R8 dataset.**

| # | Preprocessing(s) | BN | SMO | RF |
|---|---|---|---|---|
| 1 | None | 81.99 | 94.98 | 91.73 |
| 2 | C | 82.04 | 94.96 | 91.44 |
| 3 | L | 81.99 | 94.98 | 91.73 |
| 4 | P | 81.99 | 94.98 | 91.73 |
| 5 | R | 81.95 | 94.99 | 91.34 |
| 6 | S | *90.40v* | **95.75v** | 94.46v |
| 7 | CL | 82.04 | 94.96 | 91.35 |
| 8 | CP | 82.08 | 95.01 | 91.48 |
| 9 | CR | 82.01 | 94.95 | 91.24 |
| 10 | CS | 90.31v | 95.67v | *94.66v* |
| 11 | LP | 81.99 | 94.98 | 91.73 |
| 12 | LR | 81.95 | 94.99 | 91.34 |
| 13 | LS | *90.40v* | **95.75v** | 94.46v |
| 14 | PR | 81.95 | 94.99 | 91.34 |
| 15 | PS | *90.40v* | **95.75v** | 94.46v |
| 16 | SR | 90.36v | 95.71v | 94.54v |
| 17 | CLP | 82.04 | 94.96 | 91.44 |
| 18 | CLR | 82.00 | 94.96 | 91.38 |
| 19 | CLS | 90.32v | 95.66v | 94.62v |
| 20 | CPR | 81.99 | 95.00 | 91.40 |
| 21 | CPS | 90.32v | 95.68v | 94.58v |
| 22 | CRS | 90.29v | 95.68v | 94.52v |
| 23 | LPR | 81.95 | 94.99 | 91.34 |
| 24 | LPS | *90.40v* | **95.75v** | 94.46v |
| 25 | LSR | 90.36v | 95.71v | 94.54v |
| 26 | PSR | 90.36v | 95.71v | 94.54v |
| 27 | CLPR | 82.01 | 94.96 | 91.47 |
| 28 | CLPS | 90.31v | 95.67v | 94.47v |
| 29 | CLRS | 90.26v | 95.70v | 94.49v |
| 30 | CPRS | 90.27v | 95.67v | 94.40v |
| 31 | LPRS | 90.36v | 95.71v | 94.54v |
| 32 | CLPRS | 90.30v | 95.67v | 94.57v |

different from the best combination(s) for all other datasets. That is, there is no combination that turned out to be systematically the best for several datasets. For three datasets, the best combination includes more than one preprocessing method.

The S preprocessing method was the best single method for three datasets with significant improvements. However, this method was the "worst" single method for the Sentiment Labeled Sentences dataset with two significant declines and one insignificant decline. This dataset is the smallest with only 3,000 documents, which are the shortest documents with only 66 characters for each document. A possible explanation for this finding might be that the removal of the stopwords from such short documents and such small dataset adversely affects the results due to insufficient data. S was a part of the best combination only for the WebKB and SMS datasets, and a part of the 2nd best combination for the R8 datasets. In summary, the S preprocessing method has emerged as one of the most useful methods that were tested on the four examined datasets.

**Table 9. TC results for all possible combinations on the SMS spam collection dataset.**

| # | Preprocessing(s) | BN | SMO | RF |
|---|---|---|---|---|
| 1 | None | 97.40 | 97.44 | 97.62 |
| 2 | C | 97.46 | 97.40 | 97.62 |
| 3 | L | 97.36 | 97.75 | 97.74 |
| 4 | P | 96.74* | 96.83* | 97.20* |
| 5 | R | 97.44 | 97.38 | 97.68 |
| 6 | S | 98.03 | 97.11 | **98.34v** |
| 7 | CL | 97.14 | 97.76 | 97.61 |
| 8 | CP | 96.88 | 96.72* | 97.01* |
| 9 | CR | 97.47 | 97.44 | 97.67 |
| 10 | CS | 97.92 | 97.07 | 98.15 |
| 11 | LP | 98.03 | 97.11 | **98.34v** |
| 12 | LR | 96.61* | 97.11 | 97.41 |
| 13 | LS | 97.30 | 97.59 | 97.69 |
| 14 | PR | 96.76* | 96.86* | 97.24 |
| 15 | PS | 97.11 | 96.68 | 97.37 |
| 16 | SR | 97.42 | 97.29 | 97.64 |
| 17 | CLP | 96.83 | 97.18 | 97.28 |
| 18 | CLR | 97.54 | 97.57 | 97.69 |
| 19 | CLS | 97.80 | 96.87 | 98.05 |
| 20 | CPR | 97.42 | 97.39 | 97.66 |
| 21 | CPS | 97.11 | 96.53 | 97.64 |
| 22 | CRS | 97.39 | 97.44 | 97.63 |
| 23 | LPR | 97.11 | 96.68 | 97.37 |
| 24 | LPS | 97.42 | 97.29 | 97.64 |
| 25 | LSR | 96.57* | 97.10 | 97.40 |
| 26 | PSR | 97.41 | 97.29 | 97.64 |
| 27 | CLPR | 97.41 | 97.42 | 97.62 |
| 28 | CLPS | 97.23 | 96.12* | 97.83 |
| 29 | CLRS | 97.39 | 97.45 | 97.69 |
| 30 | CPRS | 97.39 | 97.43 | 97.60 |
| 31 | LPRS | 97.41 | 97.29 | 97.64 |
| 32 | CLPRS | 97.39 | 97.44 | 97.61 |

The C preprocessing method enabled insignificant improvements compared to the best baseline result in two datasets (WebKB and Sentiment), an insignificant decline for the R8 dataset, and "no change" for the SMS dataset. The C preprocessing method was a part of the best combination for three datasets including R8 although it showed an insignificant decline for this R8 as a single preprocessing method. In general, the C method was found to be an effective method that has a slight and insignificant positive impact on the TC results.

The H preprocessing method enabled an insignificant improvement compared to the best baseline result for the WebKB dataset. In addition, this method was a part of the best combination for the WebKB dataset. The H method was not activated on the other three datasets simply because their files do not contain any HTML tags.

The L preprocessing method enabled a significant improvement compared to the best baseline result in the Sentiment dataset, an insignificant improvement for the SMS dataset, an insignificant decline for the WebKB dataset, and "no change" for the R8 dataset. L was a part of the best combination for the Sentiment and SMS datasets; datasets that include the shortest

**Table 10. TC results for all possible combinations on the sentiment labelled sentences dataset.**

| # | Preprocessing(s) | BN | SMO | RF |
|---|---|---|---|---|
| 1 | None | 65.76 | 73.74 | 75.99 |
| 2 | C | 66.01 | 74.24 | 76.89 |
| 3 | L | 69.26v | 75.15 | 78.22v |
| 4 | P | 65.51 | 73.27 | 75.70 |
| 5 | R | 65.92 | 73.85 | 75.91 |
| 6 | S | 54.37* | 71.60 | 73.19* |
| 7 | CL | 69.47v | *75.81v* | **78.78v** |
| 8 | CP | 65.78 | 74.05 | 76.16 |
| 9 | CR | 66.03v | 74.18 | 76.43 |
| 10 | CS | 55.76* | 70.73 | 73.05 |
| 11 | LP | 54.37* | 71.60 | 73.19* |
| 12 | LR | 69.05v | 74.77 | 78.27v |
| 13 | LS | 69.53v | 75.20 | 78.05v |
| 14 | PR | 65.56 | 73.42 | 75.75 |
| 15 | PS | 54.79* | 70.04* | 72.56* |
| 16 | SR | 54.40* | 71.89 | 73.42* |
| 17 | CLP | 69.29v | 75.51 | 78.17 |
| 18 | CLR | *69.54v* | 75.54v | 78.37 |
| 19 | CLS | 58.12* | 73.16 | 74.96 |
| 20 | CPR | 65.75 | 74.18 | 75.92 |
| 21 | CPS | 55.51* | 70.79 | 72.76* |
| 22 | CRS | 55.76* | 70.60 | 72.60* |
| 23 | LPR | 54.79* | 70.04* | 72.56* |
| 24 | LPS | 54.40* | 71.89 | 73.42* |
| 25 | LSR | 69.18v | 75.29 | 78.00v |
| 26 | PSR | 54.81* | 70.23* | 72.94* |
| 27 | CLPR | 69.23v | 75.55 | 78.34 |
| 28 | CLPS | 57.67* | 72.79 | 75.55 |
| 29 | CLRS | 58.12* | 73.35 | 75.13 |
| 30 | CPRS | 55.45* | 70.77 | 72.65* |
| 31 | LPRS | 54.81* | 70.23* | 72.94* |
| 32 | CLPRS | 57.48* | 73.19 | 75.34 |

documents (only 82 and 66 characters per document, respectively). The L method by converting the uppercase letters into lowercase letters enables a union of words whose difference is whether the first letter is uppercase or lowercase and by that improve the quality of the spare text in these datasets. In general, the L method was found to be an effective method that has a slight and insignificant positive impact on the TC results.

The R preprocessing method enabled insignificant improvements compared to the best baseline result in two datasets (R8 and SMS), and insignificant decline for the WebKB dataset, and "no change" for the Sentiment dataset. The R preprocessing method is a part of the best combination for the R8 dataset and a part of the 2[nd] best combination for the WebKB and Sentiment datasets. Also, the R method was found to be an effective method that has a slight and insignificant positive impact on the TC results.

The P preprocessing method was the worst single preprocessing with significant declines for two datasets (WebKB and SMS), an insignificant decline for the Sentiment dataset, and "no

**Table 11. Summary of the three best accuracy results in descending order for each dataset.**

| Dataset | Our best results | | | | |
|---|---|---|---|---|---|
| | Rank | Normalizations that yielded best result | Accuracy Result | ML method that yielded best result | Baseline result |
| WebKB | 1st best | CHS | 95.74v | SMO | 94.1 |
| | 2nd best | CHRS | 95.66v | SMO | |
| | 3rd best | HS | 95.63v | SMO | |
| | best single | S | 95.61v | SMO | |
| R8 | 1st best | S, LP, PS, LPR | 95.75v | SMO | 94.98 |
| | 2nd best | SR, LPS, PSR, LPRS | 95.71v | SMO | |
| | 3rd best | CLRS | 95.70v | SMO | |
| | best single | S | 95.75v | SMO | |
| SMS Spam Collection (SMS) | 1st best | S, LP | 98.34v | RF | 97.62 |
| | 2nd best | CS | 98.15 | RF | |
| | 3rd best | CLS | 98.05 | RF | |
| | best single | S | 98.34v | RF | |
| Sentiment Labeled Sentences (Sentiment) | 1st best | CL | 78.78v | RF | 75.99 |
| | 2nd best | CLR | 78.37 | RF | |
| | 3rd best | CLPR | 78.34 | RF | |
| | best single | L | 78.22v | RF | |

change" for the R8 dataset. This finding indicates that at least for the four tested datasets, it is not recommended to remove the punctuation marks when this is the only activated preprocessing method. However, the P preprocessing method was a part of the best combination for two datasets (R8 and SMS). That is, for part of the datasets, this method combined with other methods contributes to improving the quality of TC.

We did not compare our best results to the state-of-the-art results for these datasets because of the following reasons. The state-of-the-art systems tried to achieve the best results with no restrictions on the number of features and models they used. In contrast, in all the experiments we limited our models to use only 1,000 word unigrams as explained before. Our main aim was to explore the impact of all possible combinations of five/six basic preprocessing methods on TC. Furthermore, none of the twelve previous studies investigated the four datasets we examined in this study.

## Summary, conclusions, and future work

Our main contribution and novelty is performing an extensive and systematic set of TC experiments using all possible combinations of five/six basic preprocessing methods on four benchmark text corpora (and not samples of them) using three ML methods and training and test sets. Specifically, we investigated the influence of 31 ($2^5$–1) / 63 ($2^6$–1) nonempty combinations of the following five/six basic preprocessing methods (correction of common misspelled words, conversion of uppercase letters into lowercase letters, html object removal, punctuation mark removal, stopword removal, and reduction of replicated characters). That is, we explored 31/63 combinations of preprocessing methods for each combination of dataset and ML method. We validated the proposed model on four benchmark corpora using training and test sets to

determine what preprocessing combination(s) was (were) the best suited to classification tasks, if any.

In all four datasets, we have found significant improvements compared to the baseline results. The S (stopword removal) preprocessing method was the best single preprocessing method; it was the only single preprocessing method that enabled a significant improvement for three datasets (WebKB, R8, and SMS Spam Collection). This finding is not surprising because it is not new and many TC systems apply this preprocessing method.

Regarding the best combinations for the different datasets, the C preprocessing method was a part of the best combination for three datasets, while the S (!), P, and L preprocessing methods were part of the best combination for two datasets.

The fourth dataset (Sentiment Labelled Sentences) demonstrated completely different findings to the findings of the other three datasets. The S preprocessing method presented for two ML methods significant declines. Moreover, the S preprocessing method did not appear in any of the combinations for the three ML methods that obtained significant improvements. The L preprocessing method (converting uppercase letters into lowercase letters) was the only single preprocessing method that enabled a significant improvement compared to the baseline result. The best result for this dataset was obtained using the CL combination.

The general conclusion (at least for the datasets verified) is that it is always recommended to perform an extensive and systymatic variety of preprocessing methods combined with TC experiments because it contributes to improve TC accuracy. For each of the verified datasets there was (were) different combination(s) of basic preprocessing methods that could be recommended to significantly improve the TC using a BOW representation. Therefore, the suggested advice is to experiment all possible preprocessing method combinations rather than choosing a specific combination because different combinations can be useful or harmful depending on the selected domain, dataset, ML method, and size of the BOW representation.

Future research proposals include (1) implementing TC experiments using additional feature types, e.g., word/character n-grams, skip word/character n-grams, and TF-IDF values; (2) implementing TC experiments for combinations that include more complex types of preprocessing such as lemmatization of words, expansion of common abbreviations, and using summaries of documents instead of the original documents (HaCohen-Kerner et al. [51]); (3) applying other ML methods such as deep learning methods, and (4) conducting experiments on additional benchmark corpora written in English such as the full 20 Newsgroups data, Reuters-21578, and RCV1, and corpora written in other languages.

## Supporting information

**S1 Appendix.**
(DOCX)

## Author Contributions

**Investigation:** Yaakov HaCohen-Kerner.

**Software:** Daniel Miller, Yair Yigal.

**Writing – original draft:** Yaakov HaCohen-Kerner.

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
