## [Decision Letter · Decision Letter 0]

29 Nov 2019

PONE-D-19-27170

The Influence of Preprocessing on Text Classification using a Bag-of-Words Representation

PLOS ONE

Dear Prof. HaCohen-Kerner,

Thank you for submitting your manuscript to PLOS ONE. After careful consideration, we feel that it has merit but does not fully meet PLOS ONE’s publication criteria as it currently stands. Therefore, we invite you to submit a revised version of the manuscript that addresses the points raised during the review process.

We would appreciate receiving your revised manuscript by Jan 13 2020 11:59PM. To enhance the reproducibility of your results, we recommend that if applicable you deposit your laboratory protocols in protocols.io, where a protocol can be assigned its own identifier (DOI) such that it can be cited independently in the future. For instructions see: http://journals.plos.org/plosone/s/submission-guidelines#loc-laboratory-protocols

We look forward to receiving your revised manuscript.

Kind regards,

Farhan Hassan Khan, PhD

Academic Editor

PLOS ONE

Journal Requirements:

1. Please remove your figures from within your manuscript file, leaving only the individual TIFF/EPS image files, uploaded separately.  These will be automatically included in the reviewers’ PDF.

Reviewers' comments:

Reviewer's Responses to Questions

**Comments to the Author**

1. Is the manuscript technically sound, and do the data support the conclusions?

Reviewer #1: Yes

Reviewer #2: Partly

2. Has the statistical analysis been performed appropriately and rigorously? 

Reviewer #1: Yes

Reviewer #2: I Don't Know

3. Have the authors made all data underlying the findings in their manuscript fully available?

Reviewer #1: No

Reviewer #2: Yes

4. Is the manuscript presented in an intelligible fashion and written in standard English?

Reviewer #1: Yes

Reviewer #2: Yes

5. Review Comments to the Author

Reviewer #1: The paper is very interesting and the analysis results very useful. However, some changes should be made to

improve its quality.

The novelty of the research should be clearly stated in the abstract, in the introduction and in the conclusion. I suggest better organization of the chapters.

In section 2: The references should be up-to-date, including papers published in last decade. Moreover, important papers of literature are omitted, such as Krouska, A., Troussas, C., & Virvou, M. (2016, July). The effect of preprocessing techniques on Twitter sentiment analysis. In 2016 7th International Conference on Information, Intelligence, Systems & Applications (IISA) (pp. 1-5). IEEE.

In section 3: Datasets' features should be tabulated, and examples of the corpora should be given if possible.

In section 4: The authors should state the reason why they use these algorithms for classification instead of others. A great reference for this is Krouska, A., Troussas, C., & Virvou, M. (2017). Comparative Evaluation of Algorithms for Sentiment Analysis over Social Networking Services. J. UCS, 23(8), 755-768.

The classification model should be described in text and the scheme should be redesigned.

In section 5: The outcome of the analysis is anyway quite clear and could help prospective users of these techniques.

However, the performance analysis would also benefit from the exposure of the actual confusion matrices, not only accuracy.

Moreover, it would be interesting to see a discussion of the results of the study in comparison to the results of other similar studies.

Reviewer #2: The manuscript addresses an important issue in text preprocessing since there are several available methods and a wide evaluation of some methods may be very useful for the research community and practitioners. The text is well written and generally clear, however, some points should be better discussed and evaluated.

1- It lacks information about the statistical test applied. The authors should provide more information.

2- I miss some discussion about the applied preprocessing methods, the expected impact on the quantity and quality of features and the reason for performing each one of them.

3- Looking at the datasets (third party links provided by the authors in the Data Availability Statement in the PDF file), I have some concerns about the experiments:

a. None of the datasets seems to have HTML tags to be removed.

b. In the case of WebKB, the terms are stems (they have already been preprocessed, applying Porter Stemmer).

4- It should be important to have some discussion about the impact of each preprocessing method in each dataset, such as a comparison between the raw terms and the resulting terms after the application of a single preprocessing method. What are the differences in the resultant features? How many of the 1000 unigrams are the same between the baseline and the tested preprocessing method/combination?

5- The flowchart of the experimental evaluation is not clear (Fig. 1). What is actually done in the “Re-classify” steps? I assume that: you generate new training and test sets applying the evaluated preprocessing method/combination; then a new classification model is generated from the new training set with each ML algorithm and these new models are evaluated using the new test set. Is my understanding correct? If so, the Figure must present that information for a clear understanding.

6- You performed 10 repetitions of each configuration, varying the randomly selected training and test sets. How the accuracy of those 10 repetitions vary? It would be interesting to perform some analysis of these variations and discuss it, in order to present a more complete overview of the results.

7- I miss some analysis and discussion of the results considering the differences of the datasets. I suggest presenting a Table comparing some characteristics of the dataset and discuss if the results compared to those characteristics indicates some conclusion about the proper preprocessing method for each case. Is it possible to generalize and suggest/recommend some kind of preprocessing when the dataset has some sort of characteristics? The use of more datasets may be required to complete this analysis.

8- I suggest having available a repository with all the generated training and test sets (e.g., arff files), in case someone wants to reproduce your work or test them in future works.

- Minor comments:

1- There are some alignment issues in Table 1.

2- What is the meaning of “?” in Table 1?

3- The caption of Tables 2 to 5 states “various Normalizations”, is it correct? From the text, I understood that those tables refer to the results of executions without any normalization/preprocessing method.

4- Revise the formatting of Tables 6 to 9. For example: in Table 6, the S result for RF should be in blue; there is no line separating the sets of 4 and 5 preprocessing combinations; use the same number of digits after the decimal point for all values; in Table 9, the “P” is not aligned.

5- Explain the color notation for Tables 6 to 9. Note: confirm that you can use colors in your tables since, according to the journal’s formatting rules, it seems that text color is limited to black.

6- The last sentence of Section 5 has different formatting.

7- It seems that reference 44 is not cited in the manuscript.

8- Revise the references in order to use the same style for all of them.

9- Some typos: “disambiguation of ambiguous of acronyms”, “* in indicates”, “63 (26 - 1)”

6. PLOS authors have the option to publish the peer review history of their article (what does this mean?). If published, this will include your full peer review and any attached files.

Reviewer #1: No

Reviewer #2: No

---

## [Author Response · Author response to Decision Letter 0]

10 Jan 2020

Comments to the Author

1. Is the manuscript technically sound, and do the data support the conclusions?

Reviewer #1: Yes

Reviewer #2: Partly

2. Has the statistical analysis been performed appropriately and rigorously? 

Reviewer #1: Yes

Reviewer #2: I Don't Know

3. Have the authors made all data underlying the findings in their manuscript fully available?

Reviewer #1: No

Reviewer #2: Yes

4. Is the manuscript presented in an intelligible fashion and written in standard English?

Reviewer #1: Yes

Reviewer #2: Yes

5. Review Comments to the Author

An important introduction written by the first author:

My (former) students, authors number 2 and 3, graduated our college about a year and a half ago. Although I sent them a few requests concerning the creation of confusion matrices and ARFF files they did not agree to do that.

They are currently not engaged in academic research. They are working very hard. One of them wrote me that he comes home from work very late and had about one or two hours to eat and shower. He said that since he graduated our college, he has already reformatted his computer several times and no longer has the material. The second author sent me example files from each one of the datasets and. He also wrote me that he does not have time to work on my additional requests. Anyway, we answered almost all the questions and requests. I personally spent dozens of hours in the required repairs.

Reviewer #1: The paper is very interesting and the analysis results very useful. However, some changes should be made to improve its quality.

Our response: Thank you very much.

 

The novelty of the research should be clearly stated in the abstract, in the introduction and in the conclusion.

Our response: 

Abstract - We changed and extended one of the sentences and inside this sentence, we wrote "(and this is our main research contribution)". We also added the following two sentences: The general conclusion (at least for the datasets verified) is that it is always advisable to perform a variety of preprocessing methods combined with TC experiments because it contributes to improve TC accuracy. For all the tested datasets, there was always at least one combination of basic preprocessing methods that could be recommended to significantly improve the TC using a BOW representation.

Introduction – We added, extended, and emphasized our key contributions and novelties in various sentences in the last paragraph of this Section.

Summary, Conclusions, and Future Work – At the beginning of this section, we emphasized our main contribution and novelty. Later, we extended the discussion on our general conclusion.

I suggest better organization of the chapters.

In section 2: The references should be up-to-date, including papers published in last decade. Moreover, important papers of literature are omitted, such as Krouska, A., Troussas, C., & Virvou, M. (2016, July). The effect of preprocessing techniques on Twitter sentiment analysis. In 2016 7th International Conference on Information, Intelligence, Systems & Applications (IISA) (pp. 1-5). IEEE.

Our response: In the previous version, we include summaries of a few relevant papers from the last years. At the same time, we added summaries of the following papers: 

(1) Haddi, E., Liu, X., & Shi, Y. (2013). The role of text pre-processing in sentiment analysis. Procedia Computer Science, 17, 26-32.‏

(2) Krouska, A., Troussas, C., & Virvou, M. (2016). The effect of preprocessing techniques on Twitter sentiment analysis. In 2016 7th International Conference on Information, Intelligence, Systems & Applications (IISA) (pp. 1-5). IEEE.

(3) Jianqiang, Z., & Xiaolin, G. (2017). Comparison research on text pre-processing methods on twitter sentiment analysis. IEEE Access, 5, 2870-2879.‏

In section 3: Datasets' features should be tabulated, and examples of the corpora should be given if possible.

Our response: The Datasets' features are presented in Table 2 (new table).

We add an appendix that includes two full files (as examples) from each of these four datasets.

In section 4: The authors should state the reason why they use these algorithms for classification instead of others. A great reference for this is Krouska, A., Troussas, C., & Virvou, M. (2017). Comparative Evaluation of Algorithms for Sentiment Analysis over Social Networking Services. J. UCS, 23(8), 755-768.

Our response: We added relevant material concerning the selection of appropriate ML methods from three papers:

(1) Kotsiantis, S. B., Zaharakis, I., & Pintelas, P. (2007). Supervised machine learning: A review of classification techniques. Emerging artificial intelligence applications in computer engineering, 160, 3-24.‏

(2) Fernández-Delgado, M., Cernadas, E., Barro, S., & Amorim, D. (2014). Do we need hundreds of classifiers to solve real world classification problems?. The Journal of Machine Learning Research, 15(1), 3133-3181.‏

(3) Krouska, A., Troussas, C., & Virvou, M. (2017). Comparative Evaluation of Algorithms for Sentiment Analysis over Social Networking Services. J. UCS, 23(8), 755-768.

Then, we explained why we chose the three ML methods that we applied.

The classification model should be described in text and the scheme should be redesigned.

Our response: The second reviewer asked to make the flowchart clearer and we did it. To our opinion, the enlarged flowchart includes a reasonable textual description of the model.

In section 5: The outcome of the analysis is anyway quite clear and could help prospective users of these techniques.

Our response: Thank you very much.

However, the performance analysis would also benefit from the exposure of the actual confusion matrices, not only accuracy.

Our response: As explained before, unfortunately, despite repeated requests, authors # 2 and 3 did not send me the confusion matrices.

Moreover, it would be interesting to see a discussion of the results of the study in comparison to the results of other similar studies.

Our response: (1) we wrote at the last paragraph of the "Examined Text Corpora" Section that "We did not compare our best results to the state-of-the-art results for these datasets because of the following reasons. The state-of-the-art systems tried to achieve the best results with no restrictions on the number of features and models they used. In contrast, in all the experiments we limited our models to use only 1,000 word unigrams as explained before. Our main aim was to explore the impact of all possible combinations of six basic preprocessing methods on TC." We added a new sentence: "Furthermore, none of the twelve previous studies investigated the four datasets we examined in this study." (2) However, due to your comment, we changed the title of the rightmost column in Table 1 from "Conclusions" to "Best results and conclusions" and we add for each row the best results obtained for each discussed dataset (twelve studies including three new studies). 

Reviewer #2: The manuscript addresses an important issue in text preprocessing since there are several available methods and a wide evaluation of some methods may be very useful for the research community and practitioners. The text is well written and generally clear, however, some points should be better discussed and evaluated.

Our response: Thank you very much.

1- It lacks information about the statistical test applied. The authors should provide more information.

Our response: At the end of the first paragraph of the "Experimental Results" Section, we added the following sentence "To compare the different results, we performed statistical tests using a corrected paired two-sided t-test with a confidence level of 95%."

2- I miss some discussion about the applied preprocessing methods, the expected impact on the quantity and quality of features and the reason for performing each one of them.

Our response: Regarding the applied preprocessing methods, we added a few sentences at the first paragraph of the "Model" Section that the discussed the six applied preprocessing methods and explained why we chose them. 

Regarding the expected impact of the applied ML methods (we answered to the first reviewer why we chose the applied ML methods) and the preprocessing methods on the quantity and quality of features, we mentioned in the Introduction section that the traditional model for topic-based TC is based on the bag-of-words (BOW) representation and that many ML methods have been reported to use a BOW representation of hundreds or thousands of unigram words to achieve accuracies of 90% and greater for various TC tasks. Therefore, we decided to apply the traditional and successful BOW model of 1,000 to 5,000 (in steps of 1,000) unigram words in order to check our hypothesis that the application of different combinations of preprocessing methods can improve TC results. We added a suitable paragraph at the end of the Model Section just before Fig 1 where we present a flowchart of our classification model.

3- Looking at the datasets (third party links provided by the authors in the Data Availability Statement in the PDF file), I have some concerns about the experiments:

a. None of the datasets seems to have HTML tags to be removed.

Our response: It is true that three datasets (R8, SMS Spam Collection, and Sentiment Labelled Sentences) do not contain any HTML tags. We did not know that beforehand. Consequently, we deleted in the suitable tables (tables 8-10) all the rows, which contain the H (removal of HTML tags) preprocessing method. However, the WebKB dataset (the version that we worked with) includes many documents that contain HTML tags (see for example "WebKB – Example # 2:" in the Appendix). This kind of preprocessing slightly improved the TC results for this dataset using all three ML methods.

b. In the case of WebKB, the terms are stems (they have already been preprocessed, applying Porter Stemmer).

Our response: We worked with the original WebKB, the terms in this datasets are not are only stems. We add an appendix that includes two full files (as examples) from each of these four datasets including the WebKB dataset, which we worked with. These examples show that the words were not converted to stems.

4- It should be important to have some discussion about the impact of each preprocessing method in each dataset, such as a comparison between the raw terms and the resulting terms after the application of a single preprocessing method. What are the differences in the resultant features? How many of the 1000 unigrams are the same between the baseline and the tested preprocessing method/combination?

Our response:

For each dataset (after Tables 7-10) we added a comparison concerning the impact of the single preprocessing methods. Furthermore, in Table 11, we added a row about the best single preprocessing method for each dataset. Just after Table 11, we added a paragraph summarizing the main findings for the four datasets and we added one paragraph for each single preprocessing method discussing its impact on each dataset. We also updated various sentences in the "Summary, Conclusions, and Future Work" Section. We could not provide comparison between the raw terms and the resulting terms after the application of a single preprocessing method because authors # 2 and 3 do not have the needed files and don't have time to work on it again (they are busy in their new jobs).

5- The flowchart of the experimental evaluation is not clear (Fig. 1). What is actually done in the “Re-classify” steps? I assume that: you generate new training and test sets applying the evaluated preprocessing method/combination; then a new classification model is generated from the new training set with each ML algorithm and these new models are evaluated using the new test set. Is my understanding correct? If so, the Figure must present that information for a clear understanding.

Our response: Yes. You are right. We updated two slots in the flowchart.

6- You performed 10 repetitions of each configuration, varying the randomly selected training and test sets. How the accuracy of those 10 repetitions vary? It would be interesting to perform some analysis of these variations and discuss it, in order to present a more complete overview of the results.

Our response: We are sorry. As mentioned before, my (former) students, authors number 2 and 3, graduated our college about a year and a half ago. They are not currently engaged in academic research. They are working very hard. One of them wrote me that when we work on this research he did not save the results of the 10 repetitions. We only worked with their average results. To have the detailed results for each repetition he will need to write another code (a new one) and to perform again all the experiments and he does not have time for that. 

7- I miss some analysis and discussion of the results considering the differences of the datasets. I suggest presenting a Table comparing some characteristics of the dataset and discuss if the results compared to those characteristics indicates some conclusion about the proper preprocessing method for each case. Is it possible to generalize and suggest/recommend some kind of preprocessing when the dataset has some sort of characteristics? The use of more datasets may be required to complete this analysis.

Our response: We elaborated the analysis of the results presented for each dataset (after Tables 7-10) including the impact of each preprocessing method alone. In addition, after Table 11 we added many paragraphs that include analysis concerning the analysis of the impact of each preprocessing method for all the datasets (Tables 7-10). The application of these preprocessing on additional datasets was added to the proposals for future work.

8- I suggest having available a repository with all the generated training and test sets (e.g., arff files), in case someone wants to reproduce your work or test them in future works.

Our response: As we wrote at the beginning of Section 4 " We applied three supervised ML methods: BN, SMO, and RF using the WEKA platform with their default parameters (Witten et al., 2005; Hall et al., 2009). For each TC task, we used the experimenter mode in WEKA Version 3.9.1 with the following settings: Train (67%) / Test (33%) (data randomized) and the number of repetitions of the experiment set to 10." As explained before, unfortunately, despite repeated requests, authors # 2 and 3 did not send me either ARFF or CSV files.

- Minor comments:

1- There are some alignment issues in Table 1.

Our response: Corrected.

2- What is the meaning of “?” in Table 1?

Our response: The meaning of "?" is "N/A" (i.e., not available). We changed all the appearances of “?” to "N/A".

3- The caption of Tables 2 to 5 states “various Normalizations”, is it correct? From the text, I understood that those tables refer to the results of executions without any normalization/preprocessing method.

Our response: We corrected these tables' captions. We also add the following sentence "These tables refer to the experimental results that were performed without any normalization/preprocessing methods.

4- Revise the formatting of Tables 6 to 9. For example: in Table 6, the S result for RF should be in blue; 

Our response: Please see our response to "5- Explain the color notation for Tables 6 to 9.".

… there is no line separating the sets of 4 and 5 preprocessing combinations;

Our response: Corrected.

… use the same number of digits after the decimal point for all values;

Our response: Corrected. Now, our precision is 2 digits after the decimal point for all numbers in these tables.

… in Table 9, the “P” is not aligned.

Our response: Corrected.

5- Explain the color notation for Tables 6 to 9. Note: confirm that you can use colors in your tables since, according to the journal’s formatting rules, it seems that text color is limited to black.

Our response: Corrected. All non-black colors were changed to black. In Tables 2-5, a number in blue color (that represents the best accuracy result for each ML method (one per column)) was changed to a number in italics and a number in bold blue color (the best accuracy result in the discussed table) was changed to a number in bold (see the paragraph before these tables). The same annotations and emphases have been made in Tables 6-9 and the needed changes have been performed. A suitable explanation was added in the paragraph before Tables 6-9.

6- The last sentence of Section 5 has different formatting.

Our response: Corrected.

7- It seems that reference 44 is not cited in the manuscript.

Our response: We deleted this reference.

8- Revise the references in order to use the same style for all of them.

Our response: We worked a few hours and now all the references are now written according to the same style (the APA's style). We know that the style of PLOS ONE is another one. If the paper is accepted, the style will be changed.

9- Some typos: “disambiguation of ambiguous of acronyms”, “* in indicates”, “63 (26 - 1)”

Our response: Corrected.

6. PLOS authors have the option to publish the peer review history of their article (what does this mean?). If published, this will include your full peer review and any attached files.

Do you want your identity to be public for this peer review? For information about this choice, including consent withdrawal, please see our Privacy Policy.

Reviewer #1: No

Reviewer #2: No

While revising your submission, please upload your figure files to the Preflight Analysis and Conversion Engine (PACE) digital diagnostic tool, https://pacev2.apexcovantage.com/. PACE helps ensure that figures meet PLOS requirements. To use PACE, you must first register as a user. Registration is free. Then, login and navigate to the UPLOAD tab, where you will find detailed instructions on how to use the tool. If you encounter any issues or have any questions when using PACE, please email us at figures@plos.org. Please note that Supporting Information files do not need this step. ________________________________________

---

## [Decision Letter · Decision Letter 1]

19 Mar 2020

PONE-D-19-27170R1

The Influence of Preprocessing on Text Classification using a Bag-of-Words Representation

PLOS ONE

Dear Prof. HaCohen-Kerner,

Thank you for submitting your manuscript to PLOS ONE. After careful consideration, we feel that it has merit but does not fully meet PLOS ONE’s publication criteria as it currently stands. Therefore, we invite you to submit a revised version of the manuscript that addresses the points raised during the review process.

I have read the two rounds of reviews and the authors' responses to the orignal review. 

As indicated by Reviewer #2, we can see the efforts made by the authors as well as their difficulties. 

However, there are some questions that need to be clarified and explained. 

Please refer to the reviews raised by Reviewer #2. Thanks very much.  

We would appreciate receiving your revised manuscript by May 03 2020 11:59PM. To enhance the reproducibility of your results, we recommend that if applicable you deposit your laboratory protocols in protocols.io, where a protocol can be assigned its own identifier (DOI) such that it can be cited independently in the future. For instructions see: http://journals.plos.org/plosone/s/submission-guidelines#loc-laboratory-protocols

We look forward to receiving your revised manuscript.

Kind regards,

Weinan Zhang

Academic Editor

PLOS ONE

Reviewers' comments:

Reviewer's Responses to Questions

**Comments to the Author**

1. If the authors have adequately addressed your comments raised in a previous round of review and you feel that this manuscript is now acceptable for publication, you may indicate that here to bypass the “Comments to the Author” section, enter your conflict of interest statement in the “Confidential to Editor” section, and submit your "Accept" recommendation.

Reviewer #1: All comments have been addressed

Reviewer #2: (No Response)

2. Is the manuscript technically sound, and do the data support the conclusions?

Reviewer #1: Yes

Reviewer #2: Yes

3. Has the statistical analysis been performed appropriately and rigorously? 

Reviewer #1: Yes

Reviewer #2: Yes

4. Have the authors made all data underlying the findings in their manuscript fully available?

Reviewer #1: No

Reviewer #2: No

5. Is the manuscript presented in an intelligible fashion and written in standard English?

Reviewer #1: Yes

Reviewer #2: Yes

6. Review Comments to the Author

Reviewer #1: The paper is very interesting and the analysis results very useful. The authors tackled all changes, and thus, the paper is worth for publishing.

Reviewer #2: I understand that it is difficult to continue the research when the students complete their program and start new jobs. Nevertheless, it is very important to maintain a backup of all data used/generated during the experimental evaluations.

The revision and the explanations added to the revised version of the paper improved its quality. The points that still need some attention are:

1. Re-write the last paragraph of the Introduction, describing the Sections of the paper.

2. Provide the section Materials and Methods, which were included in the revised version without content.

3. Example #1 of WebKB (Appendix) seems to be only the content of an HTML source. It does not have the HTML tags as Example #2.

4. About the R8 dataset. The examples of R8 files in the Appendix do not present upper case letters and punctuation. It seems that they are preprocessed files. Is that the case of every file of this dataset used in the experiments? Besides, in Table 8 we can see that P (removing punctuation) and L (converting to lowercase) presented the same accuracy of the baseline. This result may indicate that P and L did not make any effect since the baseline documents are already in lowercase and without punctuation. However, some of the best results for this dataset were obtained with the combination LP. How this could be explained?

5. Delete the word “twice” in the sentence “The best result in Table 10, 78.78%v was obtained twice by the RF method using CL”.

6. In the sentence “The majority of the improvement is because of the C preprocessing, which presents a significant improvement of 2.23% from the baseline”, according to Table 10, it is L instead of C.

7. PLOS authors have the option to publish the peer review history of their article (what does this mean?). If published, this will include your full peer review and any attached files.

Reviewer #1: No

Reviewer #2: No

---

## [Author Response · Author response to Decision Letter 1]

22 Mar 2020

6. Review Comments to the Author

Reviewer #1: The paper is very interesting and the analysis results very useful. The authors tackled all changes, and thus, the paper is worth for publishing.

Our response: Thank you very much for all your helpful and clever comments.

Reviewer #2: I understand that it is difficult to continue the research when the students complete their program and start new jobs. Nevertheless, it is very important to maintain a backup of all data used/generated during the experimental evaluations.

Our response: All the computers in the graduation projects lab were reset and all the files in them were deleted. We will endeavor to learn a lesson and do so in subsequent studies.

The revision and the explanations added to the revised version of the paper improved its quality. The points that still need some attention are:

Our response: Thank you very much for all your helpful and clever comments.

1. Re-write the last paragraph of the Introduction, describing the Sections of the paper.

Our response: Done.

2. Provide the section Materials and Methods, which were included in the revised version without content.

Our response: Done. 

3. Example #1 of WebKB (Appendix) seems to be only the content of an HTML source. It does not have the HTML tags as Example #2.

Our response: We worked with the original documents, which include HTML tags. We replaced the Example #1 in its original file with its HTML tags. 

4. About the R8 dataset. The examples of R8 files in the Appendix do not present upper case letters and punctuation. It seems that they are preprocessed files. Is that the case of every file of this dataset used in the experiments? Besides, in Table 8 we can see that P (removing punctuation) and L (converting to lowercase) presented the same accuracy of the baseline. This result may indicate that P and L did not make any effect since the baseline documents are already in lowercase and without punctuation. However, some of the best results for this dataset were obtained with the combination LP. How this could be explained?

Our response: This is true. The documents of R8 do not contain upper case letters and punctuation symbols. Therefore, there should be no changes in the results in Table 8 following the application of the L and P preprocessing methods. One of my students checked this matter and he wrote me that what happened is probably because the codes of several combinations of preprocessing methods in several rows of results in Table 8 have been mistakenly replaced. The document "The Influence of Preprocessing on TC-WithChanges" presents corrections, which have been made in Table 8 because of your remark. In some cases, small changes versus expected results occur due to random components.

5. Delete the word “twice” in the sentence “The best result in Table 10, 78.78%v was obtained twice by the RF method using CL”.

Our response: Corrected.

6. In the sentence “The majority of the improvement is because of the C preprocessing, which presents a significant improvement of 2.23% from the baseline”, according to Table 10, it is L instead of C.

Our response: Corrected. We changed the beginning of the relevant paragraph 

from 

"The best result in Table 10, 78.78% was obtained by the RF method using CL. The majority of the improvement is because of the C preprocessing, which presents a significant improvement of 2.23% from the baseline;"

to

"The best result in Table 10, 78.78%v (a significant improvement of 2.79% from the baseline) was obtained by the RF method using CL. The majority of the improvement is because of the L preprocessing, which presents a significant improvement of 2.23% from the baseline"

---

## [Decision Letter · Decision Letter 2]

17 Apr 2020

The Influence of Preprocessing on Text Classification using a Bag-of-Words Representation

PONE-D-19-27170R2

Dear Dr. HaCohen-Kerner,

We are pleased to inform you that your manuscript has been judged scientifically suitable for publication and will be formally accepted for publication once it complies with all outstanding technical requirements.

With kind regards,

Weinan Zhang

Academic Editor

PLOS ONE

Additional Editor Comments (optional):

Reviewers' comments:

Reviewer's Responses to Questions

**Comments to the Author**

1. If the authors have adequately addressed your comments raised in a previous round of review and you feel that this manuscript is now acceptable for publication, you may indicate that here to bypass the “Comments to the Author” section, enter your conflict of interest statement in the “Confidential to Editor” section, and submit your "Accept" recommendation.

Reviewer #1: All comments have been addressed

Reviewer #2: All comments have been addressed

2. Is the manuscript technically sound, and do the data support the conclusions?

Reviewer #1: Yes

Reviewer #2: Yes

3. Has the statistical analysis been performed appropriately and rigorously? 

Reviewer #1: Yes

Reviewer #2: Yes

4. Have the authors made all data underlying the findings in their manuscript fully available?

Reviewer #1: No

Reviewer #2: No

5. Is the manuscript presented in an intelligible fashion and written in standard English?

Reviewer #1: Yes

Reviewer #2: Yes

6. Review Comments to the Author

Reviewer #1: The authors tackled all changes improving its quality and also the topic of the paper is very interesting. Thus, the paper is worth for publishing.

Reviewer #2: All comments have been addressed.

Considering you found that several rows of results in Table 8 have been mistakenly replaced, I suppose you have double-checked the other tables as well.

Two minor comments:

- There are misplaced horizontal lines on pages 11 and 12;

- In Section Material and Methods, I suggest to explicitly describe the preprocessing methods applied, as it was done in section Model (page 17). For example, substitute “L – lowercase, H – html tags” for “L – converting uppercase letters into lowercase letters, H – HTML tag removal (for relevant datasets)”.

7. PLOS authors have the option to publish the peer review history of their article (what does this mean?). If published, this will include your full peer review and any attached files.

Reviewer #1: No

Reviewer #2: No

---

## [Editor Report · Acceptance letter]

21 Apr 2020

PONE-D-19-27170R2 

The Influence of Preprocessing on Text Classification using a Bag-of-Words Representation 

Dear Dr. HaCohen-Kerner:

I am pleased to inform you that your manuscript has been deemed suitable for publication in PLOS ONE. Congratulations! Your manuscript is now with our production department. 

With kind regards,

on behalf of

Dr. Weinan Zhang 

Academic Editor

PLOS ONE